# Insights from the care home staff on the use of observational risk assessment of contractures: Longitudinal evaluation (ORACLE): A qualitative study

Hina Tariq[1]*, Kathryn Collins[1], Joel Dunn[2], Sam Porter[1]

1 Faculty of Health and Social Sciences, Bournemouth University, Bournemouth, United Kingdom,
2 Dorset Healthcare University Foundation Trust, Dorset, United Kingdom

* htariq@bournemouth.ac.uk

## Abstract

### Background

Contractures are a common but preventable consequence of immobility and inactivity among residents living in care homes. There is a rising prevalence and subsequent impact of contractures on care home residents, leading to the development of a risk assessment tool for contractures, Observational Risk Assessment for Contractures: Longitudinal Evaluation (ORACLE). This qualitative study aims to explore the experience of care staff regarding the usability, acceptability, and practical implementation of ORACLE.

### Methods

A qualitative study using a partly deductive and pragmatic approach was conducted through semi-structured interviews with care home staff in England. The care staff members were selected via purposive sampling and were interviewed either through videoconferencing or in person in a private room at care homes. The interviews were recorded and transcribed verbatim. The data collected was coded using NVivo and synthesised using thematic analysis.

### Results

Ten care staff members were interviewed from five care homes (four senior staff members and six healthcare assistants). Three overarching themes were identified: 1) usability of ORACLE, 2) acceptability of ORACLE and 3) contextual factors that can potentially influence the practical implementation of ORACLE in a care home setting. Respondents found the tool to be user-friendly and well-integrated within existing care routines. The study also identified factors relating to care home processes, the people involved, the training environment, and the policy context that tend to support or inhibit the effective implementation of ORACLE.

**Data availability statement:** All relevant data are within the manuscript and its Supporting Information files.

**Funding:** This study was funded by Bournemouth University and Dorset HealthCare University Foundation Trust through a match-funded PhD studentship awarded to H. Tariq.

**Competing interests:** The authors have declared that no competing interests exist.

## Conclusion

The study offers preliminary insights into the usability and acceptability of ORACLE and its application in a care home setting.

---

## Introduction

Joint contractures are commonly defined as the partial or complete limitation in passive range of motion (ROM) that results from the shortening of the periarticular structures, including muscles, tendons, ligaments, joint capsules and skin spanning one or more joints [1,2]). Contractures are a common consequence of neurological conditions, e.g., stroke, Parkinson's and Alzheimer's diseases, musculoskeletal conditions like osteoarthritis and fractures and other local conditions such as burns [2–4].

The aetiology of contractures is primarily underpinned by immobility, which could be attributed to an alteration in muscle tone, decreased muscle strength, pain, decreased mobility or function, or impaired cognition [4–6]. Contractures, once developed, lead to a vicious cycle of impairments, each impacting the next, causing the progression of the original contracture [3]. When one or more joints are immobilised in a shortened position for prolonged periods, it leads to fibrotic changes within the muscles, triggering contracture development [1,7]. Initially, these contractures may be clinically non-relevant or mild; however, they increase the risk of physical impairments such as pain, discomfort, pressure sores and fractures contributing to further immobility [8]. Consequently, there is difficulty in performing activities of daily living such as eating, dressing and bathing, along with restrictions in physical mobility, increasing the need for nursing care [9–11].

In addition to the physical impairments, contractures can have a significant effect on the psychological well-being, leading to loss of autonomy, emotional distress, depressive symptoms and reduced participation in social and community activities, ultimately lowering the overall quality of life [8,10–12]. This, in turn, contributes to further deconditioning, increased dependence, increased risk of falls, progression of existing contractures and increased risk of mortality [4,13].

Joint contractures are highly prevalent among residents living in long-term care facilities. The prevalence of contractures in care home facilities ranges from 20 to 91% [10,14–16]. A study conducted by Lam et al [2022] on long-term care residents reported that a significant number of residents develop new joint contractures during the first five years of their admission to a care home [16]. Of these, 59.4% were affected in more than two limbs, while 40.5% experienced contractures involving all four limbs (arms and legs). The prevalence of contractures in the upper limbs (arms and hands) was 85.4%, which was similar to that of the prevalence in the lower limbs (legs and feet), recorded at 75% [16].

The wide variation in the prevalence of contractures could be attributed to the heterogeneous involvement of joint connective tissues in joint mobility, heterogeneous underlying conditions, the lack of a standard definition of contractures, and the lack

of a standard outcome measure for contractures [5]. The literature has highlighted that care home residents should be screened regularly to identify the risk of developing and progressing contractures [16,17].

Despite this, there has been a lack of clinical and regulatory attention to contracture prevention, unlike other areas of care, such as falls or pressure ulcers, particularly in England. The Care Quality Commission (CQC) is England's primary independent regulatory authority for adult health and social care services. Currently, care homes in England are not required to report contractures to the CQC, unlike the incidence of pressure sores, which the care homes are obliged to report. This lack of policy focus is notable when compared to other countries, where preventable joint contractures are considered a key quality indicator [11,18].

In addition to this policy gap, there is a clear lack of a standard, evidence-based, and systematic risk assessment tool for early identification of contractures and to trigger timely referrals to specialists. To address this, a contracture risk assessment tool for use in care homes, ORACLE (Observational Risk Assessment for Contractures: Longitudinal Evaluation), was recently developed [19]. This study aims to explore the experience of care home staff regarding the usability and acceptability of ORACLE and gain insights into its practical implementation.

## Methodology

This study uses a qualitative descriptive design using open-ended, semi-structured interviews [20] to explore care home staff experiences on the usability and acceptability of ORACLE and gain insights into its feasibility and practical implementation within care home settings in England. This study is nested within a multi-phase, mixed-methods research project (Trial registration: ClinicalTrials.gov NCT06042907), which aimed to develop and validate ORACLE in care homes. ORACLE (Table 1) consists of two key components: (a) a 10-item risk assessment tool designed to identify the level of risk of contractures development and progression in care home residents and (b) a response algorithm which provides guidance to the care home staff to prescribe a set of actions in response to the level of risk identified to prevent the development or progression of contractures [19]. This study conformed to the Consolidated Criteria for Reporting Qualitative Research (COREQ) [21].

### Participants and setting

The qualitative study was carried out in five care homes that participated in testing the validity and reliability of ORACLE. All participating care homes had a good rating by the CQC. Two categories of care staff were eligible to participate in the semi-structured interviews. The inclusion criteria were:

**Table 1. Overview of ORACLE.**

ORACLE comprises two main components:
**A. Risk assessment tool** which consists of the following observable indicators to calculate the risk of contractures:
• Age
• Mobility
• Muscle Weakness
• Functional ability
• Pain
• Pressure sores
• Cognition
• Activity Engagement
**B. Response Actions for Care Staff**
Based on the risk assessment findings, ORACLE provides guidance to care staff to support timely interventions such as promoting movement and independence, ensuring appropriate nutrition and hydration, reviewing pain management and medications, monitoring skin condition, and initiating specialist referrals in response to identified risk.

1) Managers, registered nurses, or senior staff members engaged in coordinating the application of ORACLE in the care homes.

2) Healthcare assistants routinely engaged in the care of residents and who conducted ORACLE assessments.

Previous research suggests that data saturation typically occurs after 6–12 interviews [22]. A purposive sample of 10 care staff members was determined, based on available resources and pragmatic considerations. Participants were purposively selected based on the eligibility criteria. The sampling process was informed by the principle of data saturation, with interviews conducted until no new themes were identified. Saturation was evaluated iteratively throughout the data analysis, and by the tenth interview, it was clear that no key themes had been repeated.

The participants included care staff working across a variety of care homes, including nursing, residential, mixed or specialised care facilities (dementia, neurological, etc.). For confidentiality purposes, participating care homes are not named. The care staff primarily cared for elderly residents with minimal support needs or those who required regular assistance with activities of daily living, individuals with cognitive impairments, learning disabilities and chronic health conditions.

### Semi-Structured interviews

The semi-structured interviews were conducted between 15th February and 30th May 2024 either remotely over secure videoconferencing or face-to-face in a private room at the care homes, depending on the convenience of the participants. The interviews lasted between 40 and 60 minutes, were audio and video recorded and transcribed verbatim by HT.

The semi-structured interview guide (S1 Appendix) was developed based on the research aims through discussion among the research team members and using results from a Delphi survey [19]. Participants were asked socio-demographic questions, including their age, gender, and years of experience in care facilities, as well as questions about the usability of ORACLE. Carers were asked about their experience of completing the ORACLE assessments, e.g., how easy or difficult it was to incorporate the tool into their work routine. Additionally, senior staff members, including managers, were asked about their experiences to gain a deeper understanding of the context of the care homes for practical implementation of ORACLE, including whether the care home had the appropriate time, staff and skills resources to perform assessments regularly. Follow-up questions were asked based on the participants' responses and experiences. Field notes were taken during and immediately after each interview to capture contextual observations.

### Ethics

This research study has been granted a favourable opinion by an independent NHS research ethics committee, Camberwell St Giles REC (IRAS Project ID: 318311) and the Research Ethics Committee at Bournemouth University (Ethics identification number: 45572). All participants provided verbal and signed informed consent at the beginning of each semi-structured interview and could opt to withdraw at any time. The data was only accessible by the research team, and all transcripts were pseudo-anonymised prior to analysis.

### Data analysis

HT checked the audio files and corresponding transcripts to correct auto-transcription errors and ensure anonymisation. The transcripts were then imported into qualitative data management and analysis software (NVivo). The transcripts were analysed using reflexive thematic analysis following Braun and Clarke's approach [23]. A coding framework was developed iteratively during analysis by KC and HT.

This was followed by a 5-step process of familiarisation, identifying a thematic framework, indexing, charting, mapping and interpretation [23]. Field notes were used to support contextualisation and interpretation of the transcripts. Coding was conducted using a hybrid approach of deductive and inductive reasoning [24]. The deductive coding was informed by

a set of a priori codes derived from the interview guide, while the inductive coding allowed for the generation of new codes and subsequent development of themes. Once coding was complete, overarching themes were developed, and the prominent quotes were identified for extraction based on those themes. Open discussions and iterative revisions were carried out among the research team to reflect on the quotes, thematic coding, and interpretation. Disagreements were resolved through discussion and consensus (HT, KC, JD, and SP).

### Researcher reflexivity

The first author, a PhD student with a physiotherapy background, conducted this study as part of their doctoral research and was closely involved in each phase of the study, from design to data collection and analysis. All participant interviews were conducted by the first author, who had prior experience and training in qualitative research methods. The broader research team, which included a physiotherapy academic, a nursing academic and a clinical physiotherapist, played an active role throughout the research from conception through to write-up. Their multidisciplinary input, coupled with regular critical discussions, helped to minimise the individual bias and enhanced the trustworthiness and methodological rigour of the study.

### Patient and public involvement and engagement (PPIE)

A face-to-face PPI session was conducted with five healthcare assistants (HCAs) from a local care home prior to the formal testing of ORACLE in care homes. They provided written and verbal feedback on the ORACLE tool, which informed the development of a short training package [25] and minor revisions in ORACLE. Their input ensured that the tool was easy to comprehend and that the terminology used was accessible for staff who lacked formal medical training.

## Results

The average length of the interviews was 40 minutes. Ten care staff members from five private care homes participated in the study. Of these, three specialised in dementia care, neurological care and learning disabilities, while two care homes offered mixed services. The maximum capacity of the care homes ranged from 17 to 69 residents.

Among participants, four were senior staff members who organised the data collection for ORACLE testing, and six were care assistants who completed ORACLE assessments for care home residents. The participants' ages ranged from 21 to 65, and their experience in the care industry ranged from 2 to 16 years. The demographic characteristics of all participants are summarised in Table 2.

Table 2. Participant demographics.

| Participant ID | Age | Gender | Job role | Speciality | Years of Experience |
|---|---|---|---|---|---|
| P1 | 46 | Female | Care Coordinator | Mixed | 5 |
| P2 | 39 | Female | Care Coordinator | Dementia | 16 |
| P3 | 44 | Male | Care home manager | Learning disabilities + Mental health | 15 |
| P4 | 39 | Female | Care Coordinator | Neurological | 4 |
| P5 | 65 | Female | Carer | Dementia | 8 |
| P6 | 28 | Female | Carer | Learning disabilities + Mental health | 2 |
| P7 | 31 | Female | Carer | Learning disabilities | 4 |
| P8 | 29 | Female | Carer | Mixed | 11 |
| P9 | 33 | Female | Carer | Dementia | 3 |
| P10 | 21 | Female | Carer | Mixed | 4 |

## Overarching themes

From the data analysis, three overarching themes were identified: 1) usability of ORACLE, 2) acceptability of ORACLE and 3) contextual factors that can potentially influence the practical implementation of ORACLE in a care home setting (Fig 1). Most of the themes and sub-themes were primarily deductive, and closely aligned with the interview guide. However, several sub-themes were developed inductively through analysis of the data. Specifically, within theme 2: acceptability, inductively derived sub-themes included *quality of care* and *behavioural changes in practice*. Within theme 3: contextual factors, inductive elements included *dementia, motivation and engagement, and consistency of care*.

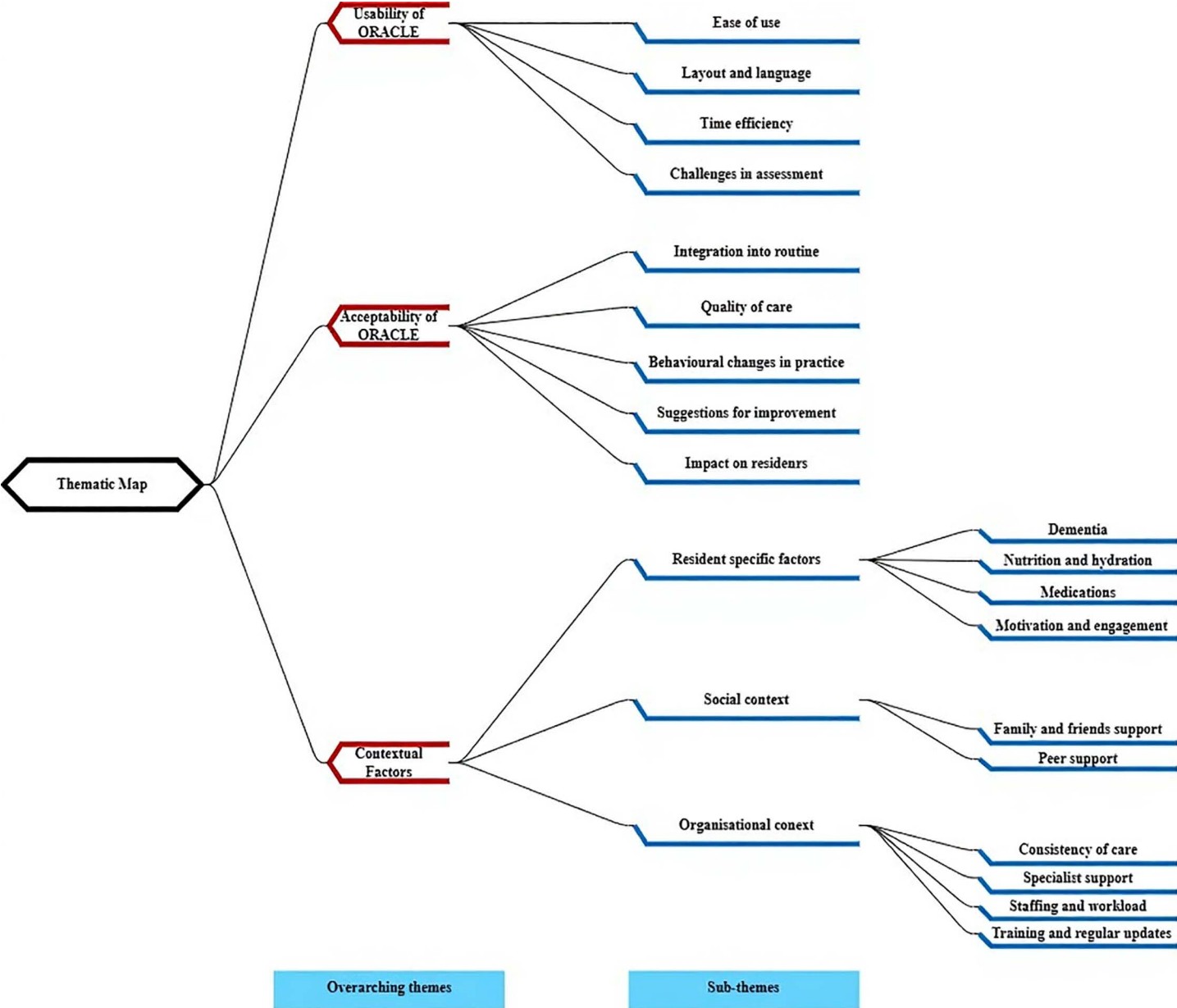

**Fig 1. Thematic map.** Fig 1 illustrates the three overarching themes and sub-themes extracted from the analysis.

**Theme 1: Usability of ORACLE.** The following sub-themes were identified within this theme:

*Usability: ease of use.* Carers consistently reported ORACLE as a user-friendly tool with a straightforward and easy-to-use scoring system. They appreciated that it allowed them to quickly sum up the scores for contracture risk assessment without complex calculations.

*'The scoring was very easy…Just add them all up, and it's straightforward' (Carer#3)*

Some senior staff members highlighted that ORACLE's simplicity and user-friendly interface enhanced its overall usability, ensuring that the tool was easily adapted by the newer staff with minimal training and exposure.

*'It was easy to understand. I think it was well set out, and easy to complete…and I think the carers were all quite comfortable using it as well…as if they'd sort of done it before, which was quite good' (Senior staff member #2)*

*Usability: layout and language.* All participants found the overall layout and design of the ORACLE simple and effective, enabling easy navigation through different sections of the tool. The use of headings, subheadings, and appropriate font size improved readability and accessibility. In addition, all participants reported that the wording used was simple, clear, and comprehensible for carers with varying levels of experience.

*'The language used was clear and easy to understand, and I liked the overall layout of this tool…yeah it was easily readable' (Carer#2)*

*Usability: time efficiency.* All carers who completed assessments on ORACLE reported that one assessment typically took 2–5 minutes to complete. One participant believed it might take a little longer for new or agency staff who are unfamiliar with the residents' overall condition.

*'Actually, it's been a really long time that I've been working here, so I know the residents here very well. So, for us, it doesn't take that long out there because we are working with them in day-to-day life. So, it only took a couple of minutes for one resident. But yeah, it might not be the same for new or agency staff' (Carer #1)*

*Usability: challenges in assessment.* Despite the overall positive reception, a few staff members found it difficult to interpret and assign predefined ranks (zero, one or two) in certain categories of ORACLE. This could potentially lead to inconsistent assessments, potentially affecting ORACLE's ability to identify the risk of contractures accurately.

Two carers specifically mentioned the category' functional ability' because some residents can perform certain activities but not others, making it difficult for them to provide a single rank.

*'Where it says if they are able to carry out activities... it was a little bit difficult to judge because some of them can eat themselves but might not be able to dress. So, I feel like maybe there could be a bit of separation or something."(Carer #5)*

*'It's really hard to put them into a zero, one, or two because some of them will dress partly, eat partly. You know, some will eat the main meal but then can't eat the pudding. It depends on the cutlery and the bowls. I struggled because some of them could dress their top half but not their lower half. So, yeah, I struggled to categorise them in that box' (Carer#5)*

One carer found assigning a rank to the category 'pain' difficult, particularly when judging residents who are unable to communicate.

*"A lot of them can't really tell you anything… so that was a bit of a judge…I was like, oh, no pain, but maybe there is pain because people with dementia can't really express it. So, that will be one that was a little bit difficult to judge."* (Carer# 6)

**Theme 2: Acceptability of ORACLE.** Acceptability in this study refers to the overall willingness of the care staff to implement ORACLE in existing work routines. It encompasses the following sub-themes:

*Acceptability: integration into routine.* Most participants reported that ORACLE assessments could be easily integrated into their care routines without significantly impacting their usual work responsibilities. One participant, however, also mentioned that it might be difficult to complete the assessment on a daily basis, but it could be easily incorporated into monthly, fortnightly, or weekly routine care assessments.

*'You're doing their charts anyway in their room… with this, you could just gauge down to check nothing's changed, and that's it. Incorporate it in when they do the daily records. And then you could do this one as well and do a risk assessment, so you'd know if anything's changed'* (Carer #3)

*'Day-to-day it would be quite difficult because every day is not the same. Sometimes we can't even spare a couple of minutes. But once a month or once every couple of weeks, it would be fine'* (Carer #4)

When asked when the best time to complete the ORACLE assessments was, most carers preferred to do so after the residents' daily care routine, whereas one carer preferred to do so in the mid-afternoon.

*'Probably after they've got up in the morning and we've done all the care…. then we could go through this to make sure if things have changed.. because we would have seen all their body…we would have moved them... we would have done all that is required and spoke to them so then that would be easy to fill out'* (Carer #1)

*'Morning and evening are quite busy for us, so anytime in mid-afternoon'* (Carer #4)

*Acceptability: quality of care.* The participants believed that ORACLE could potentially improve the quality of care of care home residents. This belief was enhanced by care home staff understanding the purpose of ORACLE, its key components, and how using it in practice can help recognise the risk associated with contractures and support care provision.

*'It covers important areas like mobility, hydration, nutrition, pain, and skin health. It all links together to give us a complete picture of the resident's risk and improve their care accordingly'* (Senior staff member #3)

*Acceptability: behavioural changes in practice.* Targeted interventions introduced in care homes, supported by staff training, improve staff confidence and the overall quality of care provided to the residents.

Carers felt that the training sessions and using ORACLE in practice helped increase their awareness of residents' mobility and risk of developing contractures. Two senior staff members also used it to train new staff members on contractures.

*'It is harder to deal with contractures than it is if they can move their legs around… because we used to have a lady, her legs were stuck…well, you can't get to do personal care…you can't do nothing because you can't force the legs… once they've developed... So, now we try and get them to at least try and move their leg. Or if they can't, we move them so that at least the muscles still moving'* (Carer #1)

*'We've had sort of recent recruitment where we've had new people come in. So I have kind of earmarked it to sort of go through with them and just give them a rough outline because I don't want to make it too complex for them'*

*(Senior staff member #1)*

*Acceptability: suggestions for improvement.* The care staff were asked for suggestions for areas of improvement in the tool. A few respondents suggested that while integrating ORACLE into routine care seems easy, it must also be carefully planned and requires a team effort to implement it effectively.

> *'I do feel it does need like a network of people to come together. I do feel it's quite a lot for, say, one or two people to do sort of independently. If a group of people, you know, get together and just work on the actual tool itself. I think that would definitely be better to apply the tool in routine care' (Senior Staff member# 4)*

Respondents also suggested regular training sessions and updates on contractures, which could improve the tool's effectiveness and the carers' familiarity with it. The agency staff should also be provided with basic training around contractures.

> *'More training, I was just about to say. Yeah, giving us more training to us would be handy' (Carer #6)*

> *'I've never seen that in any agency staff ever talk about contractures, so there's a missing link there, definitely' (Senior staff member #1)*

One carer suggested that adding a definition of contracture and/or a picture of a contracture into the tool could be beneficial.

> *'I feel adding a definition or a picture about the kind of thing would be handy' (Carer #6)*

*Acceptability: impact on residents.* Carers did not report any discomfort or negative reactions from residents during ORACLE assessments.

**Theme 3: Contextual factors.** Contextual factors are important in understanding 'what works for whom and under what circumstances' to successfully implement newly developed outcome measures in care homes [26]. Three sub-themes were developed: resident-specific factors, social context and organisational context.

*Resident-specific factors: Dementia.* Care staff found it challenging to implement the response actions of ORACLE with residents with dementia, especially those with severe cognitive deficits, as these individuals often struggle to follow instructions. They suggested that additional guidance tailored to residents with dementia might help carers provide better care. Severe cognitive impairment not only makes it difficult for residents to follow instructions but also affects their understanding of the importance of physical activity.

> *'I think that would definitely be a useful tool with some tips on how we could facilitate that with our dementia residents. Maybe that could be displayed in sort of visual cards, you know, explaining what we want to do. It's very difficult sometimes to get that engagement, you know, and get them focused on the task you want them to do'*

> *(Senior staff member #3)*

*Resident-specific factors: Nutrition and hydration.* Care staff felt that adequate nutrition and hydration are vital for maintaining energy levels for residents' mobility. While some felt overweight residents also struggle with mobility, therefore indicating a need for balanced diet management.

> *'So we had a lady [resident] diagnosed with a type of dementia which affected her perception of food, which caused her to stop eating. In the last six months, she started eating again but not full meals, only biscuits and things like that., she gained weight which affected her ability to turn as well' (Senior staff member #1)*

*'If they're not hydrated, they are going to sleep more…they're going to move less and therefore, we're doing them a disservice... and obviously then, that impacts their skin integrity. …and if their skin integrity is affected and they've got pressure sores again, they're going to be less likely wanting to move…they might be in pain' (Carer #6)*

*Resident-specific factors: Medications.* Care staff highlighted the impact of certain medications on mobility and the importance of balancing medication schedules to mitigate these risks.

*'You know, if somebody is on furosemide and they're needing the toilet a lot and they're having to get up and go, ok, they're moving… but equally, they probably just want to sleep... they're probably tired. So, they're probably sleeping, toilet sleeping, you know, and again then they might not be eating and drinking as well because they're not having a good, peaceful sleep. So yeah, it has an impact' (Senior staff member #1)*

*'…,we have residents who are on medications to help with any sort of aggressive behaviour or if they've been anxious. But it also means that they become sleepy and if it's somebody who is quite mobile, then they're at risk of falls. So, getting the timings right for their medications is important. Otherwise, if they fall, then we're going to have problems because that will, you know, have an effect on their mobility' (Senior staff member #3)*

*Resident-specific factors: Motivation and engagement.* Care staff emphasised that overall motivation and engagement in activities are critical in maintaining mobility and thus would significantly impact the effectiveness of the tool, especially for those with dementia and the elderly.

*'We have some people like that in the activities class... They just like, didn't want to eat….they don't want to drink…they don't want to do nothing' (Senior staff member #2)*

*Social context: Family/friends support.* Participants in the current study felt that family and friends' support and visits play a key role in keeping the residents motivated and active. They also shared that family members should be more involved in the care process to help the residents stay engaged and motivated.

*'You can see the families where the residents whose families come again; it gives them that sense of purpose. Again, the families are also providing their care needs, like, even if it's just, they're coming to talk to them it then it frees the carers up to be able to go and spend time with those other residents that maybe don't have the family and to be able to do the things to help them' (Senior staff member #4)*

*Social context: Peer support.* Care staff also shared that regular participation of residents in social activities at the care home keep them engaged and motivated to move around.

*'Yeah definitely, even if we're talking, we talk with our hands, don't we? And just moving… it's all the little things, even if that's anything like getting someone's attention, Oh Hello! And things like that... compared to not having any reason to move' (Senior staff member #1)*

*Organisational context: Consistency of care.* Care staff viewed that high turnover and reliance on agency staff can lead to inconsistency in care across care homes and less personalised attention for care home residents, which could be a barrier to the effective implementation of ORACLE in care homes. Moreover, as mentioned earlier, the agency staff may also struggle with ORACLE assessments due to unfamiliarity with residents.

*'.. I think in general across the industry, there is a lack of time and lack of staff…like for example we had a lady [carer] who left us, and she's gone to work in another care home, and she said, so far all I've done is work with agency staff.*

*So, there's inconsistency of care, the staff not knowing the care plans, the staff not knowing the residents' (Senior staff member #2)*

*Organisational context: Specialist support.* Most of the care staff felt that external services, such as physiotherapy and occupational therapy, are generally quick to respond to referrals.

*'I feel like it could be improved. One of the recent ones [physio] was quite good because he actually put some pictures of how he wants us to position the resident and then the pillows, and that has been very handy in comparison with other times that they've come and they just talk to the nurse. We're the ones who do their personal care, so the information gets lost, you know?' (Carer #5)*

However, some of the care assistants stressed that the specialists could be more involved in providing detailed instructions and guidance to carers so that they could perform the care plan effectively and confidently with the residents.

*'I feel like it could be improved. One of the recent ones [physio] was quite good because he actually put some pictures of how he wants us to position the resident and then the pillows, and that has been very handy in comparison with other times that they've come and they just talk to the nurse. We're the ones who do their personal care, so the information gets lost, you know?' (Carer #5)*

*'When they come, they do give us this kind of, you know, exercises to follow for some of them, but still… some people can do it. But I feel like sometimes, if am I doing it right because I don't really know what I'm doing. Even if someone explains it once to you, you know?' (Carer #2)*

*Organisational context: Staffing and workload.* Almost all care staff stressed the importance of having more staff to adequately address the needs of all residents, especially those who are less demanding but require significant attention because they are at risk of developing contractures.

*'Yeah. I think in general across the industry, there is a lack of time and lack of staff' (Senior staff member#1)*

*'If the carers are busy, you do sometimes think that some residents get more support than others… like, in our dementia unit, it's quite even. But when you go into a residential setting, the honest truth is, it's often whoever shouts the loudest. You might have a resident who's always on the buzzer, like, 'I need this, I need that.' A classic would be someone asking about dinner…like, it's in two minutes..but the carers are busy, you know?' (Senior staff member #5)*

*Organisational context: Training and regular updates.* All participants emphasised the importance of regular training on how to use the tool effectively and training on contractures, especially that includes guidance on identifying risk factors, prevention strategies, and handling dementia residents.

*'Staff need to recognise deterioration, pain, and how to safely carry out exercises' (Senior staff member #4)*

*'..they won't follow instructions, they'll just rely on you for literally everything... umm, they can't understand anything. they can't move themselves…so it's you who is doing all the jobs…. if you help exercise them, I think you could reduce a lot of the contractures' (Carer # 6)*

## Discussion

This study explored the experiences of care home staff in using the ORACLE tool, highlighting its usability, acceptability, and contextual factors influencing its practical implementation.

## Usability

Usability is a key consideration in care home settings with demanding work schedules. For successful implementation, the outcome measures must be quick to use, accessible, and easy to integrate into existing workflows [27]. Outcome measures that are time-consuming or have complex scoring systems are difficult to implement in practice and increase the chance of errors in risk assessment [27,28]. The aimed users of this tool are care staff, particularly the healthcare assistants (HCAs). Given the care home context, where formal medical assessments may not be feasible or practical for the care staff, ORACLE was developed with an emphasis on observable and physically examinable factors, which HCAs can easily and quickly assess during their usual care routines. In support of this, several studies have shown that assessment scales that are easily comprehended across different contexts and populations have broader applicability in implementation science [29].

Some challenges were identified in categorising residents into ranks using ORACLE by a few staff members. This difficulty could be attributed to the significant gap between the training provided to carers and the actual data collection, leading to potential lapses in recalling how to effectively rank the categories. Research shows that time away from training may impair skills and competence, potentially affecting the practical application of the intervention [30].

## Acceptability

Acceptability has been identified as a critical factor in the success or failure of implementing a new intervention or new practices in real-world settings, as reported by Proctor et al (2011) in their study [31].

In the current study, staff expressed confidence that ORACLE has the potential to improve the quality of care of the residents. A previous quality improvement study conducted in several care homes demonstrated that empowering care staff with education on pressure risk assessment significantly reduced the incidence of Stage I pressure ulcers [32].

Additionally, care staff reported improved awareness about the residents' mobility and contracture risk following the training sessions and the practical use of ORACLE in practice.

These findings are also consistent with those from a previous study by Petyaeva et al [2018] on the feasibility of a pain assessment intervention for care home residents with dementia. The study showed that the intervention improved staff awareness, increased staff confidence and informed decision-making across the staff [33]. Another study by Damery et al [2021] also found that upskilling care home staff can improve working practices that may be associated with reduced avoidable harms like falls, pressure ulcers, and urinary tract infections [34].

## Contextual factors

Care homes are complex environments where the implementation of an intervention or a service depends on its properties and how it interacts with the environment in which it is introduced [35]. Identification of these factors, e.g., organisational practices, provides insight into the barriers and facilitators to effective implementation and provides implementation strategies care homes can adopt to enhance the sustainability of the service introduced [26,36,37]. Three types of contextual factors were identified in this study: resident-specific, social, and organisational.

## Resident-specific factors

Care staff found it challenging to implement preventative guidance for residents with communication challenges, particularly those with severe cognitive impairments. They suggested additional guidance in ORACLE for dementia residents. Previous research has shown that using supplementary materials, such as flashcards, can improve care practices for dementia residents [33].

Care staff also highlighted the importance of nutrition and hydration in ensuring residents have the energy needed to move around. Conversely, overweight residents might also struggle with mobility. Evidence supports that both undernutrition and obesity are associated with low mobility levels among care home residents [38].

Additionally, care staff believed that certain medications and polypharmacy significantly impact residents' mobility. This is reinforced by evidence that polypharmacy is associated with mobility problems in care home residents [39]. Furthermore, adverse drug reactions, such as cognitive impairment and falls, are among the leading causes of hospital admissions in older adults [40].

### Social context

Care staff emphasised that support from family and peers plays a key role in keeping the residents socially engaged and motivating them to do physical activity, an important preventative measure for contractures. This is supported by literature, which shows a strong association between psychological and emotional well-being and physical health and disability levels in care home residents [41]. Family and peers play a key role in enhancing the residents' quality of life through visiting, offering emotional support, monitoring well-being, maintaining social connections, and promoting a positive adaptation to care home life. These factors contribute to the overall social engagement and motivation of care home residents [41,42]

### Organisational context

Care staff expressed that the guidance provided by specialists to HCAs could be more detailed and improved. Research has consistently demonstrated that high-quality external support services, like physiotherapy delivered through targeted exercise plans, can significantly benefit care home residents [43]. These services can help enhance muscle strength, improve joint flexibility, reduce pain, and promote both physical activity levels and functional independence [43,44]. Staffing levels and high workload were also identified as important organisational contextual factors in contracture prevention and effective implementation of ORACLE. Evidence suggests that staff workloads and turnover rates are high in the care home sector, which may influence the long-term sustainability of a new intervention [45]. Previous studies have reported that care staff often lack the time to integrate physical activity into daily routines due to heavy workloads [46,47]. Consequently, inadequate staff-to-resident ratios may limit the time available to implement ORACLE and related preventative measures effectively.

### Study limitations

This study had a few limitations. First, we were unable to recruit any nursing staff from the participating care homes, potentially leading to an incomplete representation of the diverse experiences of all care staff. Second, there was a time delay between the training and the actual data collection in some care homes, which could have influenced participants' recall and application of the training content. Moreover, although purposive sampling was used, the views of the participants may not be representative of all care staff. All these limitations should be considered when interpreting the findings of the study.

## Practice implications

### Care homes

Care homes that emphasise holistic care approaches, including proper nutrition and hydration, may find ORACLE more effective, as the tool's assessments are integrated into broader care strategies. Additionally, medications and polypharmacy were identified as important factors affecting the mobility levels of the care home residents. Care home staff must understand these effects and coordinate with the General Practitioner (GP) to integrate medications into the residents' routines to minimise their effect on overall mobility and improve ORACLE's efficacy.

### People

Both care assistants and residents are crucial in the practical implementation of ORACLE. Care assistants must navigate the complexities of each resident's motivation and engagement levels when using ORACLE. For example, a resident's

willingness to participate in activities or mood can significantly affect their assessment outcomes, particularly in areas related to functional ability and mobility. This also highlights the importance of understanding the residents' psychological and emotional states, often influenced by their interactions with care staff, family, and peers.

Family and friends' support is another critical factor that can influence the practical implementation of ORACLE. Residents with strong family ties may show different levels of engagement and motivation, affecting their risk of developing conditions like contractures. Family visits can either motivate residents to be more active or, conversely, lead to emotional distress that impacts their physical engagement. ORACLE's assessments must consider these dynamics, requiring care staff to make decisions that reflect the resident's physical condition and social and emotional context.

### Training and support

Given the challenges identified, particularly with language and categorisation, there is a clear need for enhanced training and support for care staff using the ORACLE tool. Providing additional guidance, such as specific guidance for Dementia residents, could help mitigate confusion and ensure consistent tool use. Organisational support is also essential, such as allocating adequate time for staff to learn and use ORACLE effectively. The findings from this study will be utilised to further refine the ORACLE and develop an implementation guide for care homes.

### Policy

Currently, the Care Quality Commission (CQC) does not explicitly mandate the prevention and management of contractures as part of its regulatory framework for care homes in England [48]. This contrasts with the Omnibus Reconciliation Act (OBRA) of 1987, which governs long-term care facilities in the United States, and explicitly states that care homes must ensure that measures are taken to prevent contractures [18]. Similarly, in Germany, joint contracture risk assessment and prevention have recently been defined as a quality indicator for nursing homes, regulated and monitored by experts from the statutory health insurance system [11]. According to this regulation, nursing homes in Germany must report whether they regularly carry out risk assessments for joint contractures and implement relevant preventative measures [11]. This highlights a critical gap in the policy that could impact the implementation of ORACLE as a standard risk assessment tool across care home settings in England. While ORACLE offers a practical solution for addressing this issue, its application in practice would be limited by the absence of a formal requirement for care homes to monitor and prevent contractures. This policy change would also ensure that care homes are held accountable for implementing effective strategies to prevent and manage contractures, thereby promoting consistency of care and improving the overall quality of care. Additionally, incorporating contracture prevention into CQC requirements would necessitate training and resource allocation changes within care homes. Staff would need to be trained in using tools like ORACLE and understanding the broader factors that contribute to or impact contracture development and prevention, such as dementia, nutrition, hydration, medication management, and psycho-social support.

### Conclusion

This study offers preliminary insights into the usability of ORACLE and its application in a care home setting. Respondents found the tool to be user-friendly and well-integrated within existing care routines. The study also identified several factors that influence the effective implementation of ORACLE, including care home processes, the individuals involved, the training environment, and the broader policy context. Successful implementation requires a comprehensive approach takes into account organisational practices, staffing levels, training, and access to specialist support. Policy frameworks and standards must also be in place to promote the application and sustainability of ORACLE and improve the overall quality of care.

## Practice implications

For care homes, key considerations should be given to organisational practices, staffing, regular training, and appropriate and timely specialist support to implement ORACLE effectively. Additionally, policy frameworks and standards must be in place to ensure accountability for care homes and the long-term sustainability of ORACLE.

## Supporting information

**S1 Appendix. Topic guide for Interviews.**
(DOCX)

**S2 Appendix. Anonymised data set.**
(DOCX)

## Acknowledgments

We would like to acknowledge the staff and management of all participating care homes for their assistance and contribution to the study.

## Author contributions

**Conceptualization:** Hina Tariq, Kathryn Collins, Sam Porter.

**Data curation:** Hina Tariq.

**Formal analysis:** Hina Tariq, Kathryn Collins.

**Funding acquisition:** Joel Dunn, Sam Porter.

**Investigation:** Hina Tariq.

**Methodology:** Hina Tariq, Kathryn Collins, Sam Porter.

**Project administration:** Hina Tariq.

**Resources:** Joel Dunn.

**Software:** Hina Tariq.

**Supervision:** Kathryn Collins, Joel Dunn, Sam Porter.

**Validation:** Kathryn Collins.

**Visualization:** Hina Tariq.

**Writing – original draft:** Hina Tariq.

**Writing – review & editing:** Hina Tariq, Kathryn Collins, Joel Dunn, Sam Porter.

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
