## [Decision Letter · Decision Letter 0]

14 Mar 2025

Dear Dr. Tariq,

We look forward to receiving your revised manuscript.

Kind regards,

Claudia Brogna

Academic Editor

PLOS ONE

“This study was funded by Bournemouth

University and Dorset Healthcare University

Foundation Trust through a match-funded PhD studentship awarded to H. Tariq”

5. We note that you have indicated that there are restrictions to data sharing for this study. PLOS only allows data to be available upon request if there are legal or ethical restrictions on sharing data publicly. For more information on unacceptable data access restrictions, please see http://journals.plos.org/plosone/s/data-availability#loc-unacceptable-data-access-restrictions .  

6. Please include captions for your Supporting Information files at the end of your manuscript, and update any in-text citations to match accordingly. Please see our Supporting Information guidelines for more information: http://journals.plos.org/plosone/s/supporting-information .

Reviewers' comments:

Reviewer's Responses to Questions

**Comments to the Author**

1. Is the manuscript technically sound, and do the data support the conclusions?

Reviewer #1: Yes

Reviewer #2: Yes

2. Has the statistical analysis been performed appropriately and rigorously?

Reviewer #1: N/A

Reviewer #2: N/A

3. Have the authors made all data underlying the findings in their manuscript fully available?

Reviewer #1: No

Reviewer #2: No

4. Is the manuscript presented in an intelligible fashion and written in standard English?

Reviewer #1: Yes

Reviewer #2: Yes

Reviewer #1: Sampling Justification: While the authors mention purposive sampling and the selection of 10 participants, a more detailed discussion on theoretical saturation and how it was determined would be beneficial.

Interview Excerpts: Some interview excerpts are included, but the results section would benefit from a richer presentation of qualitative data, incorporating more quotes to illustrate the analyses.

Discussion and Generalization: Some inferences regarding the applicability of ORACLE could be more cautious, avoiding broad generalizations based on a relatively small sample.

Data Availability: Consider providing an anonymized dataset or additional transcripts as supplementary material.

Text Flow: Minor refinements in conciseness and organization, particularly in the introduction and discussion, would enhance clarity and impact.

Reviewer #2: This is an interesting and relevant qualitative research study about the experience of care home staff of using a risk assessment tool for contractures. The study is generally well designed and methodologically sound and in parts well written. The attention to influential contextual factors is a strength. There are areas that would benefit from revision.

General:

For the most part the paper is well written however in parts there is repetition of words and points that need to be removed in a revision. For example, page 3 lines 47-54 refers to immobility and a vicious cycle twice.

Abstract:

The methods could include that this was a partly deductive and pragmatic qualitative approach and also identify which country the work was undertaken.

Introduction:

The policy issues are highlighted later in the paper - this could be part of scene setting in the introduction that in England there is lack of attention (regulatory and clinically) compared to other aspects of care prior to the point that there is lack of a standard definition and measure of contractures. It would be useful to know if this is a national or international issue and if any previous attempts to address this gap as not clear currently.

Could add more about impact on quality of life for residents in the introduction if any evidence, literature in relation to this as the emphasis tends to be more on the physical impact of contractures.

There is an assumption of clinical knowledge that not all readers may have so be beneficial to provide further explanation e.g. extremities - does this mean arms, legs, hands, feet?

Useful to have a box with more description of ORACLE either in intro when mentioned or in methodology section.

Methodology:

Add where this was - England. also info on the CQC ratings of homes - all good?

Page 6 line 102 - data saturation not theoretical saturation

Page 6 line 107-108 - state not named homes but have in the acknowledgements unless pseudo-anonymised so suggest that provide more generalised thanks especially if do not have consent from homes to be named

Semi-structured interviews section: reference needs to be made to taking field notes as referred to in the analysis section but no detail given about this in the data collection.

Data analysis: Braun & Clarke advise that themes are generated/identified - they do not 'emerge' see their recent papers on this. There is some conflation between codes and themes that needs to be addressed in this section. Were there any disagreements between the team that needed to be resolved?

A short explanation of researcher reflexivity would be beneficial.

Was there any PPIE (patient and public involvement and engagement)?

Results:

The three themes generally make sense. It could be made clearer which are more deductive/descriptive and those which are more inductive and conceptual.

The use of quotes to demonstrate the themes is generally good. Some sub-themes lack quotes which could be added. The section on acceptability and impact on residents is very brief and either needs quotes or suggest that this point is made in the description of the overall theme or in the discussion.

Some of the results text moves into discussion points and recommend that a distinct discussion section is created (see next point).

The paper lacks a discussion section and this could be created from some of references to literature in the results section and the points made in the implications so the conclusion should be a more succinct section. In the discussion can also draw on the literature from the field of implementation science for example Proctor et al 2011's taxonomy on implementation outcomes includes acceptability as an influential factor and also see Sekhon et al's 2017 acceptability theoretical framework. The importance of user experience in designing such tools could also be referenced and considered. The other issue not touched on but relevant is the increasing complexity of residents in care homes and the increasing demands on care staff and the need for external specialised support when needed. Asking more and more of an under-funded, resourced constrained care sector and workforce is a wider contextual factor that impacts implementation and useability.

Limitations: were all the staff nursing staff?

References:

Ensure all references are correct and complete as noted some are not.

Look forward to a revision of this article.

**Do you want your identity to be public for this peer review?** For information about this choice, including consent withdrawal, please see our Privacy Policy

Reviewer #1: No

Reviewer #2: No

---

## [Author Response · Author response to Decision Letter 1]

14 Jun 2025

Dear Editor,

Dear Reviewers,

We would like to thank you for the opportunity to resubmit the revised copy of the paper, “Insights from the Care Home Staff on the Use of Observational Risk Assessment of Contractures: Longitudinal Evaluation (ORACLE): A Qualitative Study”

We really appreciate your constructive and valuable feedback. We hope we have addressed your concerns to your satisfaction. Please find below our responses to each of the points raised.

Best regards ,

Hina Tariq

Response to the academic editor

We have carefully reviewed the manuscript and ensured that it meets PLOS ONE’s style requirements, including file naming. Please let us know if you find any other style violations.

Thank you for the note. This is not applicable as the manuscript does not include any author-generated code.

Thank you for pointing this out. We have reviewed and corrected the grant numbers to ensure consistency between the ‘Funding Information’ and ‘Financial Disclosure’ sections.

“This study was funded by Bournemouth

University and Dorset Healthcare University

Foundation Trust through a match-funded PhD studentship awarded to H. Tariq”

As requested, we have included the following statement in the cover letter:

5. We note that you have indicated that there are restrictions to data sharing for this study. PLOS only allows data to be available upon request if there are legal or ethical restrictions on sharing data publicly. For more information on unacceptable data access restrictions, please see http://journals.plos.org/plosone/s/data-availability#loc-unacceptable-data-access-restrictions.

Thank you for your guidance. We have now included an anonymised data set as Supporting Information to enable replication of our study findings. Full transcripts are not publicly available due to the small sample size and the inclusion of details about specific care homes, which could risk identifying participants.

We have added captions for all Supporting Information files at the end of the manuscript and updated the in-text citations accordingly.

Response to the Reviewers:

Reviewer #1 comments

1.

Sampling Justification: While the authors mention purposive sampling and the selection of 10 participants, a more detailed discussion on theoretical saturation and how it was determined would be beneficial

In the revised manuscript, we have elaborated on how saturation was determined during data collection and analysis (lines 136-140).

2. Interview Excerpts: Some interview excerpts are included, but the results section would benefit from a richer presentation of qualitative data, incorporating more quotes to illustrate the analyses.

Thank you for the valuable suggestion. We have included additional quotes to support our findings. Furthermore, an anonymised dataset has been added to the Supporting Information.

3. Discussion and Generalization: Some inferences regarding the applicability of ORACLE could be more cautious, avoiding broad generalizations based on a relatively small sample.

We have revised the discussion and conclusion to be more cautious in tone regarding the broader applicability of ORACLE.

4. Data Availability: Consider providing an anonymized dataset or additional transcripts as supplementary material.

Selected anonymised excerpts from participant interviews are included in the Supplementary Information. Full transcripts are not publicly available due to the small sample size and the inclusion of details about specific care homes, which could risk identifying participants.

5. Text Flow: Minor refinements in conciseness and organization, particularly in the introduction and discussion, would enhance clarity and impact.

We have reviewed the introduction and discussion sections and made revisions to improve conciseness and clarity.

Reviewer #2 comments

1. General:

For the most part the paper is well written however in parts there is repetition of words and points that need to be removed in a revision. For example, page 3 lines 47-54 refers to immobility and a vicious cycle twice.

This has been reviewed and corrected.

2. Abstract:

The methods could include that this was a partly deductive and pragmatic qualitative approach and also identify which country the work was undertaken.

We have updated the abstract (lines 20–21) as per your suggestion.

3. Introduction:

The policy issues are highlighted later in the paper - this could be part of scene setting in the introduction that in England there is lack of attention (regulatory and clinically) compared to other aspects of care prior to the point that there is lack of a standard definition and measure of contractures. It would be useful to know if this is a national or international issue and if any previous attempts to address this gap as not clear currently.

We have revised the introduction (lines 95–101) to include the policy context.

4. Could add more about impact on quality of life for residents in the introduction if any evidence, literature in relation to this as the emphasis tends to be more on the physical impact of contractures.

We have revised the introduction (lines 69–74) to include evidence on the impact of contractures on residents’ psychological well-being and quality of life, to provide a more holistic context.

5. There is an assumption of clinical knowledge that not all readers may have so be beneficial to provide further explanation e.g. extremities - does this mean arms, legs, hands, feet?

Thank you for highlighting this. We have revised the text (lines 82–85) to replace “extremities” with the clearer term “limbs” (arms, hands, legs, and feet) to enhance understanding for all readers.

6. Useful to have a box with more description of ORACLE either in intro when mentioned or in methodology section.

We have added a descriptive box (Table 1) for ORACLE in the methodology section.

7. Methodology:

Add where this was - England. also info on the CQC ratings of homes - all good?

We have specified the study was conducted in England (line 114) and added that all participating care homes had good CQC ratings (line 126).

Page 6 line 102 - data saturation not theoretical saturation

This has been corrected.

Page 6 line 107-108 - state not named homes but have in the acknowledgements unless pseudo-anonymised so suggest that provide more generalised thanks especially if do not have consent from homes to be named

We have revised the acknowledgement section to provide a generalised thanks to the care homes (lines 680-681).

8. Semi-structured interviews section: reference needs to be made to taking field notes as referred to in the analysis section but no detail given about this in the data collection.

We have added a brief description of taking field notes during interviews in Semi-Structured interviews section to align with the analysis details (line 163-165).

9. Data analysis: Braun & Clarke advise that themes are generated/identified - they do not 'emerge' see their recent papers on this. There is some conflation between codes and themes that needs to be addressed in this section. Were there any disagreements between the team that needed to be resolved?

We have made changes throughout the paper to align with Braun & Clarke’s guidance on theme generation/identification and to clarify the distinction between codes and themes. Additionally, we have added how team disagreements were resolved (lines 190-191).

10. A short explanation of researcher reflexivity would be beneficial.

We have added a short explanation of researcher reflexivity to the methodology section (lines 192-200).

11. Was there any PPIE (patient and public involvement and engagement)?

Thank you for highlighting this. We have added a brief section on Patient and Public Involvement and Engagement (PPIE) to the methodology section (lines 201-207).

12. Results:

The three themes generally make sense. It could be made clearer which are more deductive/descriptive and those which are more inductive and conceptual.

We have clarified within the Results section which themes are more deductive and which are inductive (lines 223-228).

13. The use of quotes to demonstrate the themes is generally good. Some sub-themes lack quotes which could be added.

We have added the supporting quotes to the sub-themes that lacked them.

14. The section on acceptability and impact on residents is very brief and either needs quotes or suggest that this point is made in the description of the overall theme or in the discussion.

Thank you for your comment. Participants were asked to report any negative impact on residents during their ORACLE observations, which was a yes/no question; therefore no proper quotations are available for this sub-theme.

15. The paper lacks a discussion section and this could be created from some of references to literature in the results section and the points made in the implications so the conclusion should be a more succinct section.

Thank you for the suggestion. We have drafted a separate Discussion section by drawing on the literature cited in the results and your recommendations. We have also revised the conclusion to be more succinct.

16. Limitations: were all the staff nursing staff?

Thank you for your question. No nursing staff were available for interview, which we have acknowledged as a limitation. The participants were either senior staff members (either care managers or care coordinators) who organised the ORACLE assessments or Healthcare assistants (HCAs) who carried out the assessments.

17. References:

Ensure all references are correct and complete as noted some are not.

We have reviewed and corrected all references to ensure they are complete and accurate.

---

## [Decision Letter · Decision Letter 1]

14 Oct 2025

Insights from the Care Home Staff on the Use of Observational Risk Assessment of Contractures: Longitudinal Evaluation (ORACLE): A Qualitative Study

PONE-D-24-50857R1

Dear Dr. Tariq,

We’re pleased to inform you that your manuscript has been judged scientifically suitable for publication and will be formally accepted for publication once it meets all outstanding technical requirements.

Kind regards,

Van Thanh Tien Nguyen, Ph.D.

Academic Editor

PLOS ONE

Additional Editor Comments (optional):

Reviewers' comments:

Reviewer's Responses to Questions

**Comments to the Author**

Reviewer #1: All comments have been addressed

Reviewer #2: All comments have been addressed

2. Is the manuscript technically sound, and do the data support the conclusions?

Reviewer #1: Yes

Reviewer #2: Yes

3. Has the statistical analysis been performed appropriately and rigorously?

Reviewer #1: Yes

Reviewer #2: N/A

4. Have the authors made all data underlying the findings in their manuscript fully available?

Reviewer #1: Yes

Reviewer #2: Yes

5. Is the manuscript presented in an intelligible fashion and written in standard English?

Reviewer #1: Yes

Reviewer #2: Yes

Reviewer #1: The manuscript presents a rigorous and well-structured qualitative approach, investigating the applicability of the ORACLE tool in the care practices of long-term care institutions for the elderly. The authors’ response to the reviewers was thorough, respectful, and careful, demonstrating a genuine effort to address the recommendations. The revisions made substantially strengthen the manuscript, enhancing both its scientific content and its formal compliance with PLOS ONE journal standards.

Reviewer #2: Thank you for the response to comments and great to see the improvements to the paper in the revised version. This is an important topic for policy stakeholders to be aware of regarding the impact of contractures for people living in care homes in England, how staff can support people and noting the need for specialist support to care home setting that is sensitive to that context.

**Do you want your identity to be public for this peer review?** For information about this choice, including consent withdrawal, please see our Privacy Policy

Reviewer #1: No

Reviewer #2: No

---

## [Editor Report · Acceptance letter]

PONE-D-24-50857R1

PLOS ONE

Dear Dr. Tariq,

I'm pleased to inform you that your manuscript has been deemed suitable for publication in PLOS ONE. Congratulations! Your manuscript is now being handed over to our production team.

Kind regards,

on behalf of

Asst. Prof. Van Thanh Tien Nguyen

Academic Editor

PLOS ONE